# A Lightweight Anchor-Based Incremental Framework for Multi-view Clustering

Anonymous

## ABSTRACT

The rapid development of multi-media techniques boosts the emergence of multi-view data, and how to uncover its intrinsic structure and utilize it to conduct the subsequent downstream tasks is crucial in data analysis. Multi-view clustering is representative of handling multi-view data. The anchor-based method has received widespread attention for excellent performance and low time complexity. However, existing methods encounter two drawbacks, cutting down their performance, i.e., the assumption of the availability of all views and limited interaction of anchor generation among views. In some scenes, views arrive sequentially, and storing them is challenging owing to the limited space/privacy considerations, and the existing anchor-based MVC is unsuitable for this. Additionally, recent works fail to generate anchors with the guidance of other views, and it is tough to align the anchor graphs. To this end, we propose A Lightweight Anchor-Based Incremental Framework for Multi-view Clustering. Specifically, we first initialize an anchor graph with the assistance of $k$-means when a new view arrives. Then, the consensus one of the anchor graph is updated by the newly collected view with a permutation matrix. Our proposed method is more capable of anchor alignment because, in incremental MVC, the anchor graphs of previous views could be listed as a reference to guide the generation of anchor graphs of the coming view. Furthermore, we design a three-step iterative and convergent algorithm to address the resultant problem. Notably, the proposed algorithm shows outstanding effectiveness and time/space efficiency in extensive experiments.

## CCS CONCEPTS

• **Theory of computation → Unsupervised learning and clustering**; • **Computing methodologies → Cluster analysis**.

## KEYWORDS

multi-view clustering, incremental learning, anchor-based method

## 1 INTRODUCTION

Cluster analysis, a fundamental scientific problem in machine learning, is widely used in data mining. The rapid development of multi-media techniques boosts the emergence of multi-view data, and multi-view clustering (MVC) is a representative of handling multi-view data by uncovering its intrinsic structure and discovering

*ACM MM, 2024, Melbourne, Australia*
© 2024 Copyright held by the owner/author(s). Publication rights licensed to ACM.
ACM ISBN 978-x-xxxx-xxxx-x/YY/MM
https://doi.org/10.1145/nnnnnnn.nnnnnnn

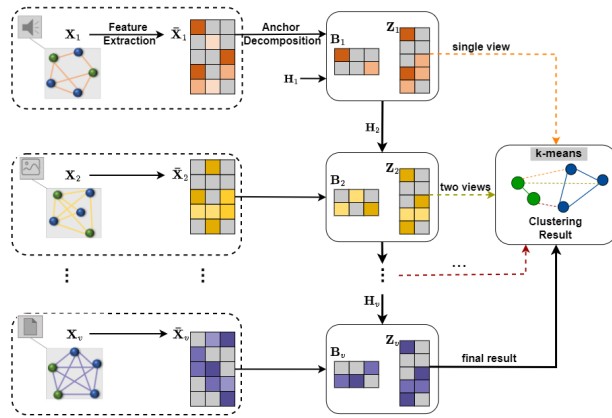

**Figure 1: Overview of the proposed LAIMVC. We store and update a consensus anchor graph to avoid fusing the anchor graphs repeatedly. After the arrival of a new view, we first perform feature extraction on $X_t$ to obtain lightweight data $\bar{X}_t$. Then we update the anchor graph with the assistance of a rotation matrix.**

proper clusters. The core of MVC is to integrate and reconcile the information from different views to reveal more comprehensive insights than a single view alone [2, 7, 15, 16, 27].

As MVC approaches achieving great success in clustering theory, existing research in MVC can be roughly classified into four categories, i.e., multi-view subspace clustering (MVSC), multiple kernel clustering (MKC), multi-view graph clustering (MVGC), and multi-view clustering based on matrix factorization (MVCMF). Specifically, MVSC generates the inherent consistency and complementarity of the data into consistent low-dimensional subspace and performs eigen-decomposition on the corresponding subspace [4, 25, 29–31]. MKC learns a consensus kernel by following a multi-kernel learning framework in which the individual kernels of each view are combined and conduct clustering on the consensus kernel[11, 13, 23, 34]. Most MVGC algorithms construct the similarity graph for each view and perform graph fusion and spectral clustering to obtain optimal unified graph and clustering results [10, 22]. As described in [5], MVCMF transforms multi-view data into data matrices and reveals their underlying consensus structure through matrix techniques such as feature decomposition of the data matrices, getting the final result on the consensus matrix subsequently[3, 9, 32].

Nevertheless, the prominent drawback of the above methods is the high time complexity consumption, such as cubic complexity in subspace/graph construction and the spectral clustering process, which hinders their application to large-scale scenarios. To handle this handicap, anchor-based MVC has been proposed to alleviate the high complexity of traditional methods [6, 17, 19]. Specifically, it

builds a consensus graph by sampling anchors to capture consistent spatial structure across multiple views. The time complexity of it is reduced to linear concerning sample number per iteration. However, existing anchor-based methods encounter two drawbacks, which are the assumption of the availability of all views and the limited interaction of anchor generation among views, leading to their performance decreasing. In some scenes, views arrive sequentially, and storing them is challenging owing to privacy considerations/the limited space, such as the multi-angle videos allowed to be obtained within a specific time range, thus the existing anchor-based MVC is unsuitable. Additionally, recent works fail to generate anchors with the guidance of other views, causing difficulty in the alignment of the unified anchor graph.

To address aforementioned issues, we propose a novel algorithm termed A **L**ightweight **A**nchor-Based **I**ncremental Framework to **M**ulti-**v**iew **C**lustering (LAIMVC). It combines incremental learning and the anchor-based method into a unified framework. The illustration of our framework is shown in Fig. 1. Specifically, LAIMVC effectively harmonizes distinct views by reducing dimensions via Principal Component Analysis (PCA) to facilitate the subsequent anchor graph fusion. We generate the anchor graph of the first view by conducting $k$-means on it. Then, the consensus one of the anchor graphs is updated by the newly collected view with a permutation matrix, which not only avoids the repetitive computation of sampling anchors but enhances the interaction of anchor generation among views. Especially, the algorithm not only has linear time complexity for large-scale multi-view data, but it can cope well with view-sequential data costing only linear space complexity. What's more, our proposed method is more capable of anchor alignment because, in incremental MVC, the anchor graphs of previous views could be listed as a reference to guide the generation of anchor graphs of the coming view. Admittedly, a three-step alternative strategy is proposed to solve the optimization problem and the convergence can be guaranteed. Experimental results on seven multiple datasets show that the method can obtain comparable or better performance.

The main contributions of our proposed framework can be listed as follows,

(1) LAIMVC is a creative attempt to extend anchor-based MVC to an incremental framework. By merely retaining the consensus anchor matrix and graph pertaining to all prior views, it iteratively integrates views individually until no further novel views come, making it applicable to sequential-view data scenarios.

(2) Different from existing methods that align anchor graphs trickly, LAIMVC alleviates this problem owing to the guidance of the previous consensus anchor graph. Thus it increases the clustering performance and efficiency of the learned anchor graph structure.

(3) An alternate optimization method with proven convergence is developed to address the resulting issue. LAIMVC shows clearly superior clustering performance and significantly less running time/storing space compared with state-of-the-art methods, so it is critical for large-scale and sequential-view datasets, especially with limited computing sources.

## 2 RELATED WORK

In this section, we will introduce some related researches, including continual multi-view clustering and multi-view anchor graph clustering.

### 2.1 Continual Multi-view Clustering

Continual Multi-view Clustering (CMVC), proposed by Wan et al. [21], combines continual learning with the late fusion method. CMVC maintains a consensus partition matrix and further enhances it by accumulating new views. The base partition matrix utilizes a permutation matrix to align old and new information appropriately. Additionally, it introduces a regularization parameter to balance the weights of two sets of information effectively.

When the $t$-view arrives, with $H_t$ as the base partition and the consensus partition matrix $H^*_{t-1}$ maintained, the optimal permutation matrix can be computed by maximizing the objective formulation in Eq. (1).

$$\max_{\widetilde{H}_t, W_t} \text{Tr}\left(\widetilde{H}_t^\top H_t W_t\right) + \lambda \text{Tr}\left(\widetilde{H}_t^\top H^*_{t-1}\right),$$
$$\text{s.t. } \widetilde{H}_t^\top \widetilde{H}_t = I_k, W_t^\top W_t = I_k. \tag{1}$$

As indicated in Eq.(1), the update process only requires two matrices, making it more adaptable and time-saving. Maintaining the consensus partition matrix instead of individual base partition matrices reduces the storage space needed for previous views. Overall, the CMVC approach is more flexible and cost-effective in terms of both time and space.

### 2.2 Multi-view Anchor Graph Clustering

Multi-view anchor graph clustering has been proposed to make multi-view clustering methods suitable for large-scale data. Usually, traditional graph clustering approaches attach $O(vn^2)$ space and $O(n^3)$ time complexity to instance numbers [18], while multi-view anchor graph clustering can effectively reduce the time and space consumption to $O(vnm)$ and $O(nm^2)$ by constructing anchor graphs on $v$ views and $m$ anchors instead of the all $n$ instances.

Constructing the anchor graph of each view is the first step in the multi-view anchor graph clustering. The following equation can implement this process,

$$\min_{Z_i} \|X_i - Z_i A_i\|_F^2 + \beta \|Z_i\|_F^2, \text{ s.t. } Z_i \geq 0, Z_i 1_m = 1_n. \tag{2}$$

The matrix $A_i \in \mathbb{R}^{m \times d_i}$ serves as the anchor matrix for the $i$-th view. The matrix $Z_i \in \mathbb{R}^{n \times m}$ represents the anchor graph for the $i$-th view, and the equation $Z_i 1_m = 1_n$ ensures that the similarity of each instance to all anchors is standardized. It should be noted that the quality of anchors significantly impacts clustering effectiveness, along with some literature on anchor sampling methods.

After constructing the view-specific anchor graphs $\{Z_i\}_{i=1}^v$, the final anchor graph $G \in \mathbb{R}^{n \times m}$ can be fused as follows,

$$\min_{\alpha, G} \left\|\sum_{i=1}^v \alpha_i Z_i - G\right\|_F^2, \text{ s.t. } G1_m = 1_n, \alpha^\top 1_v = 1, \alpha \geq 0. \tag{3}$$

The weight coefficient $\alpha \in \mathbb{R}^v$ measures the different impact of each view, and the constraint $\alpha^\top 1_v = 1$ ensures that $G1_m = 1_n$. The optimization process alternates between the weight coefficient

$\alpha$ and fused graph $\mathbf{G}$ within a unified framework. In a follow-up study, the assumptions of equal weights for all views and the rank constraints on the fusion anchor graph are proposed to ensure a non-trivial solution [6, 8]. After obtaining the fusion graph, k-means clustering is applied to its left singular vectors for final clustering.

## 3 METHOD

In this section, we present the formulation of LAIMVC and then propose a three-step alternate optimization approach to address the issue. Following that, we provide its convergence, complexity, and potential expansion.

### 3.1 Formulation

In light of challenges faced by prevailing anchor-based multi-view clustering (MVC) methods when dealing with sequentially arriving or storage-restricted views, we propose LAIMVC to creatively extend anchor-based MVC into an incremental framework. Unlike prior works that struggle with generating anchors without cross-view guidance and encounter difficulties in aligning a unified anchor graph, LAIMVC addresses these limitations effectively.

Given a multi-view data matrix $\mathbf{X}_t$ of the $t$-th view with $n$ samples, $d_t$ denotes its feature dimension. Considering that the inconsistent of dimensions among views might lead to the difficulty in anchor graph integration, we first conduct PCA on the feature matrix to a unified dimension $d$ as $\widetilde{\mathbf{X}}_t$ when the $t$-th view is collected. And the consensus anchor graph $\mathbf{Z}_t \in \mathbb{R}^{n \times m}$ of previous views has already been attained, under its guidance, we update the consistent anchor graph in an incremental framework to avoid the alignment of anchor graphs among views in the literature. $\Omega$ is a regularization term. We formulate it as follows,

$$
\min_{\mathbf{Z}_t, \mathbf{B}_t} \left\| \widetilde{\mathbf{X}}_t - \mathbf{Z}_t \mathbf{B}_t \right\|_{\mathbf{F}}^2 + \Omega(\mathbf{Z}_t, \mathbf{Z}_{t-1}, \mathbf{B}_t, \mathbf{B}_{t-1}),
$$
$$
\text{s.t. } \mathbf{Z}_t^\top \mathbf{Z}_t = \mathbf{I}_m, \mathbf{B}_t^\top \mathbf{B}_t = \mathbf{I}_d. \tag{4}
$$

A small trick is that a permutation matrix $\mathbf{H}_t \in \mathbb{R}^{m \times m}$ is imposed to allow the change of the anchor graph from a certain degree. As shown in Eq.(5), the permutation matrix ensures the sustainability of data decomposition while performing data alignment. When $\mathbf{H}_t \mathbf{H}_t^\top = \mathbf{I}_m$ is satisfied, we have

$$
\widetilde{\mathbf{X}}_{t-1} = \mathbf{Z}_{t-1} \mathbf{B}_{t-1} = \mathbf{Z}_{t-1} \mathbf{H}_t \mathbf{H}_t^\top \mathbf{B}_{t-1}. \tag{5}
$$

By imposing the square of the Frobenius norm as regularization with the trick in Eq.(5), we can achieve the optimal anchor graph by minimizing the specific objective formulation as,

$$
\min_{\mathbf{Z}_t, \mathbf{B}_t, \mathbf{H}_t} \left\| \widetilde{\mathbf{X}}_t - \mathbf{Z}_t \mathbf{B}_t \right\|_{\mathbf{F}}^2 + \left\| \mathbf{Z}_t - \mathbf{Z}_{t-1} \mathbf{H}_t \right\|_{\mathbf{F}}^2 + \left\| \mathbf{B}_t - \mathbf{H}_t^\top \mathbf{B}_{t-1} \right\|_{\mathbf{F}}^2,
$$
$$
\text{s.t. } \mathbf{Z}_t^\top \mathbf{Z}_t = \mathbf{I}_m, \mathbf{B}_t^\top \mathbf{B}_t = \mathbf{I}_d, \mathbf{H}_t^\top \mathbf{H}_t = \mathbf{I}_m. \tag{6}
$$

As shown in Eq.(6), LAIMVC merely retains the consensus anchor matrix and associated anchor graph pertaining to all prior views, eschewing the save of base matrices or anchor graphs specific to each view. Besides, upon acquisition of a new view, the consensus anchors and graph are jointly refined with a permutation matrix, which fosters interaction in graph generation across views. It iteratively integrates views individually until all views are

attained, we ultimately derive the final clustering results through $k$-means on the last consensus graph $\mathbf{Z}_v$.

### 3.2 Alternate Optimization

Eq.(6) comprises three distinct variables requiring optimization, a task that can prove arduous when performed concurrently. So we develop a three-step alternate optimization, optimizing one variable while the rest remain fixed. The formula detailed derivation is given in the appendix.

**Updating $\mathbf{Z}_t$.** Fixing $\mathbf{B}_t$ and $\mathbf{H}_t$, the optimization in Eq.(6) w.r.t $\mathbf{Z}_t$ can be reduced to

$$
\max_{\mathbf{Z}_t} \text{Tr} \left( \mathbf{Z}_t^\top \mathbf{A} \right), \text{ s.t. } \mathbf{Z}_t^\top \mathbf{Z}_t = \mathbf{I}_m, \tag{7}
$$

where $\mathbf{A} = \widetilde{\mathbf{X}}_t \mathbf{B}_t^\top + \mathbf{Z}_{t-1} \mathbf{H}_t$. The optimum of $\mathbf{Z}_t$ can be analytically obtained with rank $m$ truncated the singular value decomposition (SVD) by [24]. Specifically, if the SVD form is $\mathbf{A} = \mathbf{S}\Sigma\mathbf{V}^\top$, the closed-form solution in Eq.(7) is as follows

$$
\mathbf{Z}_t = \mathbf{S}\mathbf{V}^\top. \tag{8}
$$

**Updating $\mathbf{B}_t$.** Fixing $\mathbf{Z}_t$ and $\mathbf{H}_t$, the optimization w.r.t $\mathbf{B}_t$ can be reduced to

$$
\max_{\mathbf{B}_t} \text{Tr} \left( \mathbf{B}_t^\top \mathbf{C} \right), \text{ s.t. } \mathbf{B}_t^\top \mathbf{B}_t = \mathbf{I}_d, \tag{9}
$$

where $\mathbf{C} = \mathbf{Z}_t^\top \widetilde{\mathbf{X}}_t + \mathbf{H}_t^\top \mathbf{B}_{t-1}$. Similar to Eq.(7) and suppose the SVD form is $\mathbf{C} = \mathbf{S}'\Sigma\mathbf{V}'^\top$, the closed-form solution in 9 is as follows

$$
\mathbf{B}_t = \mathbf{S}'\mathbf{V}'^\top. \tag{10}
$$

**Updating $\mathbf{H}_t$.** Fixing $\mathbf{Z}_t$ and $\mathbf{B}_t$, the optimization w.r.t $\mathbf{H}_t$ can be reduced to

$$
\max_{\mathbf{H}_t} \text{Tr} \left( \mathbf{H}_t^\top \mathbf{D} \right), \text{ s.t. } \mathbf{H}_t^\top \mathbf{H}_t = \mathbf{I}_m, \tag{11}
$$

where $\mathbf{D} = \mathbf{Z}_{t-1}^\top \mathbf{Z}_t + \mathbf{B}_{t-1} \mathbf{B}_t$. Similar to Eq.(7) and suppose the SVD form is $\mathbf{D} = \mathbf{S}''\Sigma\mathbf{V}''^\top$, the closed-form solution in 11 is as follows

$$
\mathbf{H}_t = \mathbf{S}''\mathbf{V}''^\top. \tag{12}
$$

The optimization process of the LAIMVC is outlined in Algorithm 1. It can be seen that the views arrive sequentially so the $\widetilde{\mathbf{X}}_t$ and $\mathbf{Z}_t^t$ are generated when $t$-th view is obtained. The convergence of LAIMVC in theory and its time/space complexity and potential expansion will be argued in the following part.

### 3.3 Discussion

*3.3.1 Convergence.* In this section, we give the theoretical proof of the convergence of the proposed LAIMVC, which is the important support for the solvability of the optimization algorithm.

In Eq.(6), every square term is not less than 0. Therefore, the objective function has a lower bound. We will verify in the following part the objective value of Eq.(6) monotonically decreases along with iterations. For ease of expression, we simplify the objective formula as,

$$
\min_{\mathbf{Z}_t, \mathbf{B}_t, \mathbf{H}_t} \Gamma(\mathbf{Z}_t, \mathbf{B}_t, \mathbf{H}_t). \tag{13}
$$

As demonstrated in Algorithm 1, the optimization process consists of three iterative parts every iteration, i.e. $\mathbf{Z}_t$ subproblem, $\mathbf{B}_t$ subproblem and $\mathbf{H}_t$ subproblem. Suppose the superscript $i$ indicate the optimization process at $i$-th round. The convergence analysis is given as follows.

**Algorithm 1** A **L**ightweight **A**nchor-Based **I**ncremental Framework for **M**ulti-**V**iew **C**lustering

---

**Input:** $\{X_t\}_{t=1}^{v}$, $d$, $m$, cluster number $k$ and $\varepsilon_0$.
**Output:** $Z_v$.
1: **for** $t = 1$ to $v$ **do**
2:     Preprocessing: $X_t$ to $\widetilde{X}_t$ by PCA.
3:     **if** $t = 1$ **then**
4:        Initialization: $\widetilde{X}_1 = Z_1 B_1$ by $k$-means and $H_1 = I_m$.
5:     **else**
6:        $i = 1$
7:        **while** not converged **do**
8:           Update $Z_t$ by solving Eq.(7).
9:           Update $B_t$ by solving Eq.(9).
10:          Update $H_t$ by solving Eq.(11).
11:          $i \leftarrow i + 1$
12:        **end while**$\left(obj^{i-1} - obj^i\right)/obj^i \leq \varepsilon_0$
13:     **end if**
14: **end for**

---

1) $Z_t$-subproblem: Given $B_t^{(i)}$ and $H_t^{(i)}$, $Z_t^{(i+1)}$ can be updated via Eq.(7), so

$$\Gamma(Z_t^{(i+1)}, B_t^{(i)}, H_t^{(i)}) \leq \Gamma(Z_t^{(i)}, B_t^{(i)}, H_t^{(i)}). \tag{14}$$

2) $B_t$-subproblem: Given $Z_t^{(i+1)}$ and $H_t^{(i)}$, $B_t^{(i+1)}$ can be updated via Eq.(9), so

$$\Gamma(Z_t^{(i+1)}, B_t^{(i+1)}, H_t^{(i)}) \leq \Gamma(Z_t^{(i+1)}, B_t^{(i)}, H_t^{(i)}). \tag{15}$$

3) $H_t$-subproblem: Given $Z_t^{(i+1)}$ and $B_t^{(i+1)}$, $H_t^{(i+1)}$ can be updated via Eq.(11), so

$$\Gamma(Z_t^{(i+1)}, B_t^{(i+1)}, H_t^{(i+1)}) \leq \Gamma(Z_t^{(i+1)}, B_t^{(i+1)}, H_t^{(i)}). \tag{16}$$

Combining Eq.(14), Eq.(15) and Eq.(16), we know,

$$\Gamma(Z_t^{(i+1)}, B_t^{(i+1)}, H_t^{(i+1)}) \leq \Gamma(Z_t^{(i)}, B_t^{(i)}, H_t^{(i)}), \tag{17}$$

which illustrates that the objective value monotonically decreases with iterations. Together with the objective function existing in a lower bound, the algorithm is theoretically convergent. Furthermore, we will verify the convergence of it in the experiment.

*3.3.2 Complexity Analysis.* In this part, we will analyze the space and time complexity of our algorithm.

    **Space Complexity:** In our paper, the major memory costs of LAIMVC are matrices $Z_t$ and $B_t$. Thus the space complexity of our LAIMVC is $O(nm + md)$. In our algorithm, $m \ll n$ and $d \ll n$. Therefore, the space complexity is $O(n)$.

    **Time Complexity:** The computational complexity of LAIMVC is composed of three steps as mentioned before. Preprocessing costs $O(nd^2 + d^3)$. When updating $Z_t$, $B_t$ and $H_t$ for one iteration, it costs $O(nm^2 + mk^2 + m^3)$. Let $T$ denote the maximum number of iterations and $V$ represent the number of views, the time complexity of LAIMVC is $O(nd^2 + d^3 + TV(nm^2 + mk^2 + m^3))$, which is linear complexity with respect to $n$.

    In conclusion, LAIMVC achieves MVC with both linear space and time complexity, which demonstrates efficiency.

*3.3.3 Potential Expansion.* To the best of our knowledge, LAIMVC combines incremental learning and anchor graph clustering for the first time, which expands application to sequential scenarios and will provide inspiration for future research. Furthermore, the proposed LAIMVC is more flexible and efficient by incremental learning of anchor points and graphs than the traditional anchor-based methods. The proposed framework has the potential to be used in more complicated data scenarios or extended to other MVC paradigms due to its simple operation and scalability.

**Table 1: Information of the datasets in our experiments.**

| Dataset | Samples | Views | Clusters |
|---|---|---|---|
| Dermatology | 358 | 2 | 6 |
| Flower17 | 1360 | 7 | 17 |
| Cora | 2708 | 4 | 7 |
| AWA10 | 5814 | 6 | 10 |
| ALOI | 10800 | 4 | 100 |
| YouTubeFace10 | 38654 | 4 | 10 |
| YouTubeFace100 | 195537 | 4 | 100 |

## 4 EXPERIMENTS

In this section, we conduct experiments to compare LAIMVC with several state-of-the-art multi-view clustering methods, including three incremental, three anchor-based, and two other methods on several representative datasets. After that, intrinsic structure, running time and complexity comparison, view-fusion performance, parameter sensitivity, and convergence are discussed.

### 4.1 Experimental Settings

*4.1.1 Datasets.* Seven benchmark datasets of different categories are tested to prove the effectiveness of LAIMVC, including Dermatology[1], Flower17[2], Cora[3], AWA10[4], ALOI[5], YouTubuFace10[6], YouTubeFace100[7]. The brief information on each dataset is summarized in Table 1 and the details are given in the appendix. The first four datasets are regular datasets and the last three are large-scale datasets.

*4.1.2 Compared Algorithms and Experimental Setup.*

   (1) **OPLFMVC**[14] is a pattern of late fusion multi-view clustering using the one-pass method to attain a discrete partition matrix.
   (2) **OPMC**[12] is a one-pass algorithm that successfully unifies the multi-view matrix factorization with partition generation.
   (3) **LMVSC**[6] implements spectral clustering on the final anchor graph learned by the anchor graph for each view, which handles large-scale data with linear complexity.

---

[1]https://www.kaggle.com/datasets/syslogg/dermatology-dataset/data
[2]https://www.robots.ox.ac.uk/Ëœvgg/data/flowers/17/
[3]https://relational.fit.cvut.cz/dataset/CORA
[4]https://cvml.ista.ac.at/AwA2/
[5]https://www.kaggle.com/alvations/aloi-dataset
[6]https://www.cs.tau.ac.il/Ëœwolf/ytfaces/
[7]https://www.cs.tau.ac.il/Ëœwolf/ytfaces/

**Table 2: Empirical evaluation and comparison of LAIMVC with eight baseline methods on seven benchmark datasets in terms of clustering accuracy (ACC), normalized mutual information (NMI), and Purity, individually. Noting that 'OM' means out of the CPU memory.**

| Datasets | OPLF-MVC | OPMC | LMVSC | FPMVS-CAG | SMVSC | IMSC | SCGL | CMVC | Proposed |
|---|---|---|---|---|---|---|---|---|---|
| ACC(%) | | | | | | | | | |
| Dermatology | 74.30 | 92.46 | 78.33 | 78.10 | 79.85 | 45.81 | 89.21 | 70.39 | **96.65** |
| Flower17 | 30.74 | 33.09 | 34.95 | 26.03 | 28.36 | 34.85 | 29.19 | 36.10 | **39.56** |
| Cora | 30.98 | 40.51 | 40.02 | 39.72 | 41.88 | 33.97 | 37.11 | 34.45 | **56.68** |
| AWA10 | 24.70 | 25.56 | 24.17 | 24.35 | 25.83 | 22.14 | 24.70 | 20.88 | **26.47** |
| ALOI | 1.20 | 61.04 | 51.11 | 31.87 | 33.35 | 49.49 | 46.71 | 5.67 | **72.35** |
| YouTubeFace10 | 70.30 | 85.95 | 73.89 | 67.15 | 66.91 | 61.18 | OM | 82.29 | **93.60** |
| YouTubeFace100 | OM | 59.68 | 63.52 | 49.12 | 56.79 | OM | OM | 62.64 | **66.48** |
| NMI(%) | | | | | | | | | |
| Dermatology | 75.09 | 90.06 | 69.58 | 81.71 | 74.52 | 62.70 | 80.97 | 61.00 | **92.15** |
| Flower17 | 30.26 | 31.50 | 31.70 | 26.03 | 27.75 | 32.04 | 29.46 | 25.29 | **35.50** |
| Cora | 18.00 | 15.78 | 23.15 | 19.64 | 21.96 | 10.65 | 30.95 | 16.29 | **32.51** |
| AWA10 | 10.21 | **11.83** | 9.61 | 10.40 | 11.30 | 5.72 | 9.40 | 6.13 | 10.55 |
| ALOI | 2.07 | 78.94 | 66.70 | 63.56 | 60.12 | 69.78 | 38.40 | 15.77 | **83.18** |
| YouTubeFace10 | 77.20 | 83.37 | 76.66 | 73.27 | 74.35 | 65.16 | OM | 83.88 | **88.26** |
| YouTubeFace100 | OM | 82.23 | 81.38 | 72.82 | 78.81 | OM | OM | 74.17 | **82.70** |
| Purity(%) | | | | | | | | | |
| Dermatology | 81.01 | 94.69 | 80.51 | 79.34 | 82.26 | 67.32 | **99.31** | 76.54 | 96.65 |
| Flower17 | 31.54 | 34.93 | 33.74 | 27.42 | 29.11 | 39.34 | 30.29 | 29.41 | **39.56** |
| Cora | 38.81 | 41.14 | 46.42 | 44.93 | 43.66 | 39.29 | 52.95 | 41.03 | **59.68** |
| AWA10 | 26.63 | 29.00 | 27.03 | 27.06 | 28.46 | 24.01 | 26.63 | 23.82 | **30.75** |
| ALOI | 1.92 | 62.69 | 52.29 | 32.46 | 34.43 | 55.68 | 59.98 | 6.27 | **73.91** |
| YouTubeFace10 | 77.48 | 86.95 | 76.74 | 69.99 | 68.81 | 64.65 | OM | 86.37 | **93.60** |
| YouTubeFace100 | OM | 73.28 | 70.03 | 50.32 | 58.60 | OM | OM | 67.25 | **74.59** |

**Table 3: Running time in seconds of different algorithms on seven benchmark datasets. 'OM' means the time is not measured because of out of the CPU memory.**

| Datasets | OPLF-MVC | OPMC | LMVSC | FPMVS-CAG | SMVSC | IMSC | SCGL | CMVC | Proposed |
|---|---|---|---|---|---|---|---|---|---|
| Dermatology | 0.1042 | 0.0095 | 0.7964 | 3.2699 | 2.5809 | 0.1546 | 0.1661 | 0.0111 | 0.0078 |
| Flower17 | 0.5152 | 26.7161 | 8.9177 | 68.7341 | 75.3058 | 0.1655 | 11.4447 | 0.0026 | 0.0061 |
| Cora | 0.4409 | 19.2164 | 12.1254 | 44.3189 | 29.7203 | 0.5272 | 14.8226 | 0.0427 | 0.0105 |
| AWA-10 | 3.2870 | 31.6476 | 55.6577 | 109.9325 | 138.9711 | 1.3385 | 149.5615 | 0.0909 | 0.0083 |
| ALOI | 81.4702 | 3.6072 | 103.5729 | 131.382 | 743.2308 | 16.7888 | 611.6307 | 0.3328 | 0.0632 |
| YouTubeFace10 | 47.7445 | 14.4631 | 378.2295 | 520.4237 | 572.0479 | 341.231 | OM | 0.3920 | 1.5629 |
| YouTubeFace100 | OM | 348.7288 | 8758.3464 | 11073.1700 | 17372.6258 | OM | OM | 61.0345 | 12.0038 |

(4) **FPMVS-CAG**[26] proposes a subspace clustering method that jointly conducts anchor selection and subspace graph construction without any extra hyper-parameters.

(5) **SMVSC**[20] proposes a scalable multi-view subspace clustering with unified anchor points, which combines anchor learning and graph construction into a unified framework.

(6) **IMSC**[33] implements multi-view spectral clustering by storing several consensus graphs of previous views and updating them when a new view is collected.

(7) **SCGL**[28] proposes an efficient incremental multi-view clustering method by seeking a consensus sparse and connected graph.

(8) **CMVC**[21] combines continual learning and late fusion into a unified framework, whose time complexity is linear concerning sample number.

We obtained the public code for all the algorithms mentioned above from their original websites. The parameters for the comparison methods were set based on recommendations from the relevant literature. In the experiment, the number of anchor points in LAIMVC is chosen within $[k, 5k]$. It is assumed that for all datasets, the true number of clusters is known and set to be equal to the true number of classes.

The clustering performance is evaluated using commonly used metrics including clustering accuracy (ACC), normalized mutual information (NMI), and Purity. To mitigate the impact of randomness caused by $k$-means, each experiment is repeated 20 times with random initialization, and the average result is reported for all algorithms. Our experiments are conducted on a PC with Intel(R) Core(TM)-i7-6800K 3.4GHz CPU and 96G RAM environment. Moreover, the code of LAIMVC will be available on GitHub after acceptance.

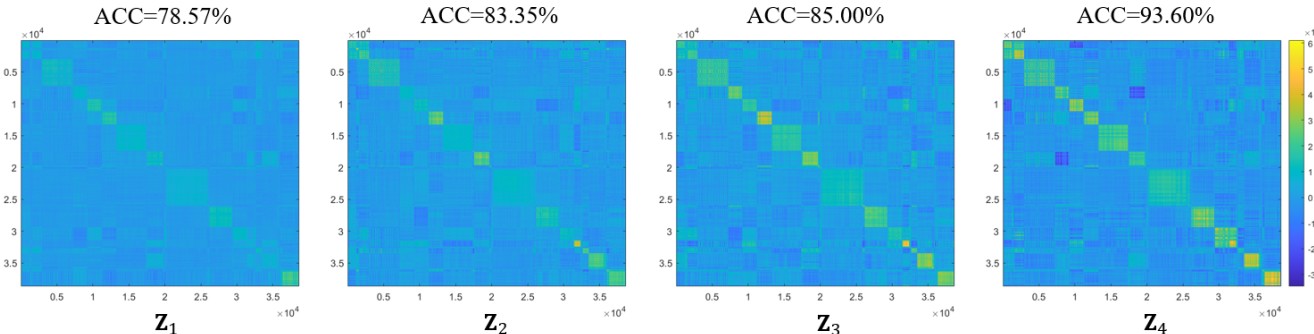

**Figure 2: The complete graphs display with 4 views incrementally on YouTubeFace10. It verifies the effectiveness of the LAIMVC in capturing the intrinsic structure of the multi-view dataset.**

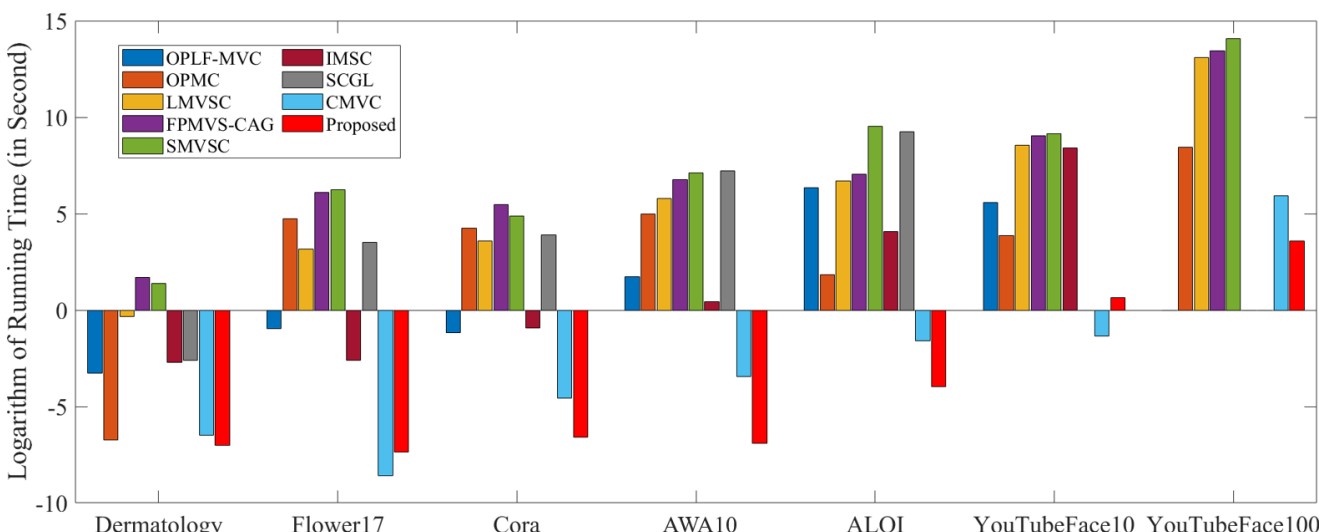

**Figure 3: The running time comparison of our proposed algorithm and the competitors on seven benchmark datasets. Noting that missing bars indicate that the methods encounter the out-of-memory error on our experimental platform under this dataset.**

## 4.2 Clustering Performance Comparison

Table 2 compares the ACC, NMI, and Purity of the aforementioned clustering methods. The notation "OM" signifies that certain algorithms could not produce results due to memory constraints. For instance, when applied to datasets containing over 30,000 samples, SCGL consumes tens of thousands of computational units per parameter tuning and subsequently encounters the issue. And we can illustrate the following conclusions:

(1) On four small-scale datasets with fewer than 10,000 samples, LAIMVC achieves top scores on three evaluation metrics in the majority of cases. Taking the results on Cora as an instance and comparing to the algorithm with the second-best results, LAIMVC outperforms the SMVSC by 14.8% in ACC and exceeds the SCGL by 1.56% for NMI and 6.73% for Purity.

(2) For the remaining three large-scale datasets exceeding 10,000 samples, LAIMVC displays a remarkable edge. During the experiments, the regular incremental MVC algorithms, except for the CMVC that solves large-scale data, suffer directly from the "out-of-memory" problem. Our proposed LAIMVC even performs better on these large datasets, more than the second method by 11.31%, 4.24%, and 11.22% on ACC, NMI, and Purity, respectively on ALOI.

In summary, as an incremental multi-view clustering technique, LAIMVC can handle scenarios where the number of views is variable, significantly enhancing the performance over anchor-based counterparts. overall, the proposed method can nearly achieve the best clustering performance on listed metrics among the compared algorithms.

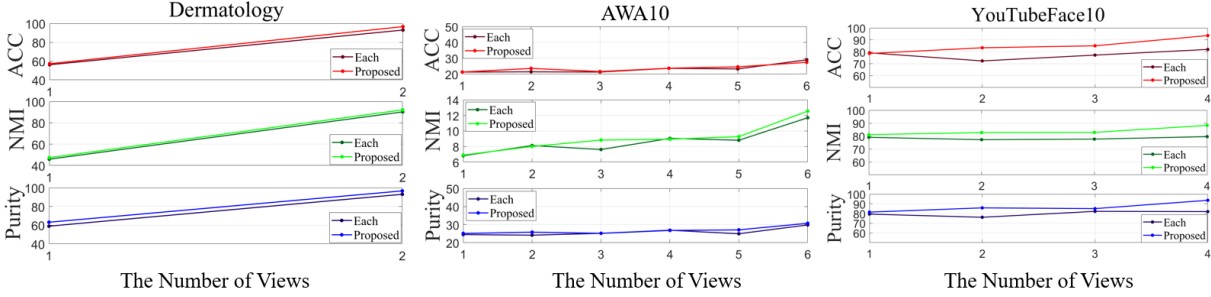

**Figure 4: The comparison of clustering performance between representation by view-fusion of LAIMVC and each view without other information.**

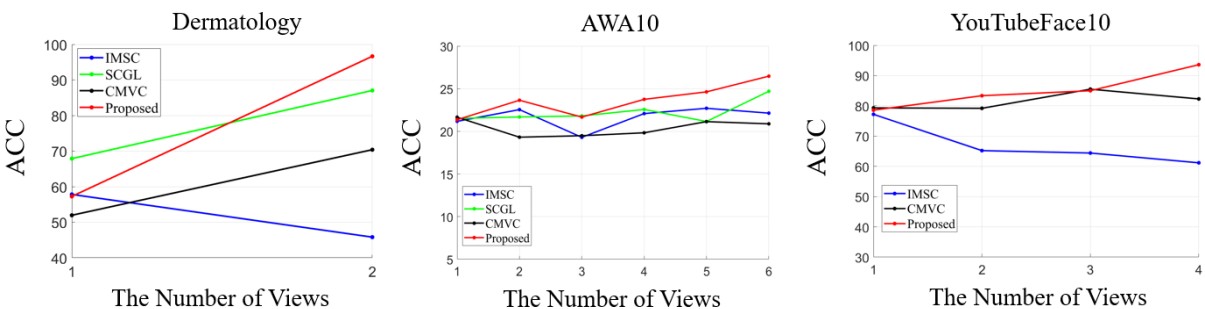

**Figure 5: The view-fusion performance in ACC with views collected in order on three datasets of LAIMVC and other three incremental multi-view clustering methods.**

### 4.3 Intrinsic Structure

Since the anchor graph $Z_t$ of each view in LAIMVC is learned with the new view coming, we construct the corresponding complete graphs by $Z_t Z_t^\top$. The complete graphs can be used to show the relationships between pairs of samples and the existing intrinsic structure on a certain dataset.

In LAIMVC, the anchor graph for previous views is continuously updated upon introducing new views, we accordingly reconstruct the associated complete graphs. For instance, we plot complete graphs of YouTubuFace10 in Fig. 2. As the number of views increases, the complete graph shows a clearer block structure with an improved clustering result, demonstrating the incremental learning capability of our proposed method in capturing the intrinsic structure and illustrating it as suitable for sequential-view scenarios and a good blend of integrating knowledge among different views.

### 4.4 Running Time and Complexity Comparison

We conduct experiments to assess the time effectiveness of LAIMVC, whose results are displayed in Table 3. It should be noted that the execution time of LAIMVC is from initialization to the final fusion of views, while other incremental algorithms exhibit similar calculations. Non-incremental algorithms record the time from the start of processing to obtaining the final characteristic representation of the data. We also plot the log-processed running time in Fig. 3 to make it intuitive. It can be concluded that LAIMVC, since it combines the time advantages of anchor graph decomposition and incremental method, has a much faster running time than other

algorithms in processing multi-view data, surpassing the current mainstream rapid clustering One-Pass method.

The time complexity can be corresponded to the running time written above. As for the space complexity, we compare the time complexity and storage space required by our method with the other methods, as detailed in the appendix. To summarise, LAIMVC has the lowest time and space complexity, together with CMVC. Especially when the number of views $v$ increases sharply, LAIMVC will show significant advantages.

In conclusion, our LAIMVC algorithm has a lower computing complexity handling data, realizes superior clustering performance on most datasets, and can be used in situations where the number of views can increase over time. We hope that the effectiveness and excellent efficiency of LAIMVC make it a not-bad consideration for applications in practical clustering scenarios and boost improvement in clustering theory.

### 4.5 View-Fusion Performance

Over time, previous views may not be available due to privacy concerns or memory restrictions. Most existing methods may be disturbed which makes it difficult to utilize the consistent and complementary information during view fusion, affecting the clustering performance. To solve the problem of dynamic collection of data views, the proposed incremental framework preserves a consensus representation of all previously acquired data and uses it to guide the view-fusion.

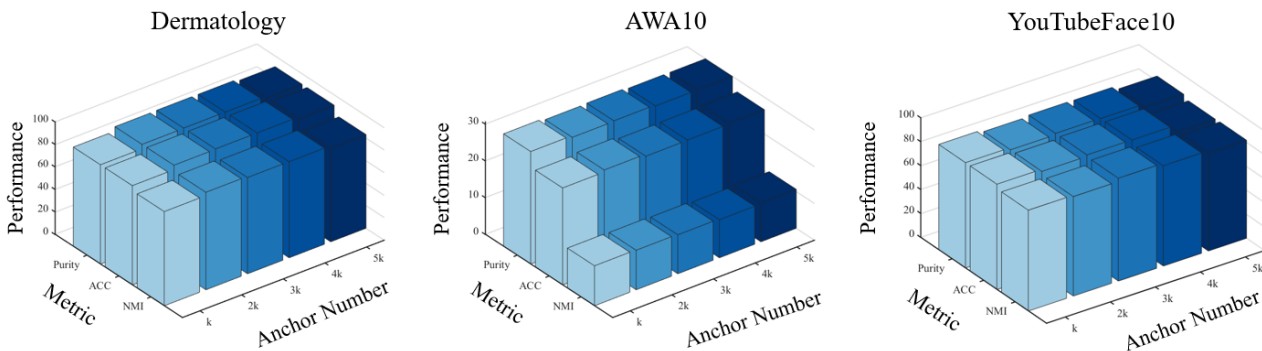

**Figure 6: The sensitivity analysis of LAIMVC with the variation of anchor number on Dermatology, AWA10, YouTubeFace10, respectively.**

To show the view-fusion performance, we carry out experiments where the views are sequentially accumulated over time. We plot the results of views integrated one by one in Fig. 4 and Fig. 5 across three distinct datasets, which are Dermatology, AWA10, and YouTubeFace arranged in ascending order of sample number.

In Fig. 4, 'Each' denotes the clustering performance on each view without any other process. Generally speaking, all three metrics are appreciably improved proving that the attempt to utilize previous information to guide the clustering process is vital. And, our proposed method can reduce the influence of view quality on clustering results to some extent.

In Fig. 5, we compare the view-fusion performance of our method with existing incremental multi-view clustering methods. The results of the other two metrics are similar and given in the appendix. With the increase in the number of views, our proposed method maintains satisfactory clustering effects and shows excellent sustainable learning ability. For instance, on AWA10 and YouTube-Face10, the clustering performances of the compared methods show downward trends when dealing with the last obtained views, while LAIMVC still shows a gain effect. Meanwhile, our proposed method remains a promising performance even though a view arrives with poor clustering quality, such as on AWA10.

Through these visual representations, we demonstrate how our method progressively improves the clustering performance as more views are combined. The detailed experiments on the effect of the preprocessing of LAIMVC on low-quality views are presented in the appendix, which is omitted here due to space limitations.

### 4.6 Parameter Sensitivity

In our method, we tune the number of anchor points according to the true number of classes $k$. So we conduct a sensitivity experiment about the anchor number to illustrate the impact of which on performance. As shown in Fig. 6, when the anchor number varies from $k$ to $5k$, it is observed that the values of representative metrics generally remain within a stable range. This indicates that the clustering performance can remain stable when the number of anchor points varies within a wide range, which also illustrates the superior robustness of our proposed method.

### 4.7 Convergence

According to [1], our algorithm is theoretically guaranteed to converge to a local minimum. The examples of the objective values with iteration are shown in Fig. 7 on Dermatology and YouTube-Face10 datasets (one regular dataset and one large-scale dataset). The results on other datasets or other views are similar and omitted due to space limits. It can be clearly verified that its objective value decreases monotonically and the algorithm converges in less than 15 iterations.

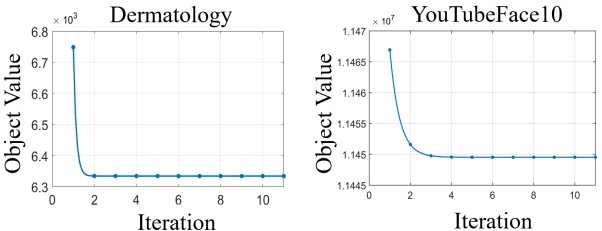

**Figure 7: The objective values of LAIMVC when fusing one view. For the fixed dataset, the target value changes when a new view is processed, and we randomly choose one procedure to visualize, while the rest effects are similar.**

### 5 CONCLUSION

In this paper, we propose a method to solve the clustering problem for sequential-view data efficiently termed a lightweight anchor-based incremental framework for multi-view clustering (LAIMVC). On the one hand, the theoretical analysis of LAIMVC with incremental learning and anchor graph theory is provided, including the convergence and linear space and time complexity. On the other hand, extensive experiments exhibit the superiority of our proposed method, including the intrinsic structure, running time and complexity comparison, view-fusion performance, parameter sensitivity, and convergence. In the future, we intend to explore the framework for the scenario of incremental sample numbers or incomplete data, etc.

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
