# OpenReview forum: "An Lightweight  Anchor-Based Incremental Framework to Multi-view Clustering"
_acmmm.org/ACMMM/2024/Conference — MM2024 Poster_

### Official Review · Reviewer_9b31 · 2024-05-07

**Rating:** 6
**Confidence:** 4

**Summary:**

Existing multi-view anchor-based clustering methods never consider incremental scenarios, resulting in limited application. The paper extends anchor-based MVC to an incremental framework, while handling the difficulty of the alignment for anchor graphs. Extensive experiments on different datasets demonstrate the effectiveness and efficiency of the proposed method.

**Strengths:**

1. The paper is well-motivated. The existing anchor-based methods ignore the dynamic learning scenario, it is the first attempt to extend anchor-based MVC to an incremental framework.
2. The experiments demonstrate its effectiveness well.
3. The authors provide the code and enhance its reproducibility.
4. The complexity analysis of the method is given, and the authors compare it with existing methods in appendix to show its superiority.

**Limitations:**

1. In the algorithm, the reason for choosing PCA as the preprocessing method could be explained. Also, other dimension-conduction methods should be compared to demonstrate its superiority.
2. The view order might have a significant effect on the performance. The authors should investigate this in experiments.
3. The novelty of Eq.(6) should be claimed. It seems that the equation is widespread in the literature.
4. Pay more attention to the layout details. For example, both Figure 4 and Figure 5 are three parallel discount graphs, taking care to keep their figure and font size consistent.
5. There is no notation table in the main body, the authors could provide it for readability.
6. In Table 2, the bold fonts should be clarified.

**Suitability:**

3

---

### Official Review · Reviewer_gtyd · 2024-05-21

**Rating:** 5
**Confidence:** 3

**Summary:**

This paper presents a novel algorithm, LAIMVC, designed to address the challenges of multi-view clustering in scenarios where data views are sequentially arriving or storage is restricted. LAIMVC is an incremental learning approach that integrates anchor-based methods, using a permutation matrix to enhance anchor alignment and interaction across views. This paper provides not only theoretical analysis but also extensive experiment results of the algorithm's effectiveness.

**Strengths:**

1. Innovative Motivation: LAIMVC combines incremental learning with anchor-based clustering, offering a new perspective for handling sequential-view data, which is a significant contribution to the relevant field.
2. Comprehensive Experiments: Experimental results show that LAIMVC outperforms the performance of state-of-the-art methods, which demonstrates the effectiveness of the proposed method.
3. Scalability: LAIMVC can handle sequentially arriving views without storing all views, making it highly scalable and applicable to real-world data streams.

**Limitations:**

1. Necessary Question: How does LAIMVC algorithm perform better than other incremental multi-view clustering methods in terms of time and space complexity? In which cases are its advantages more obvious?
2. Clarity and Conciseness of Expression: The expression might not be clear or accurate enough.        The last paragraph of the introduction section appears to be somewhat verbose, which should be tightened to enhance. There may be misinterpretation or confusion about accurately conveying complex ideas in English. Please pay attention to polish.
3. Reference Format: There are some questions in the format of references. For example, references 29 and 30 are repeated, and the title format of references 32 and 33 is not standardized. Please pay attention to modification.

**Suitability:**

3

---

### Official Review · Reviewer_s15R · 2024-05-24

**Rating:** 5
**Confidence:** 2

**Summary:**

The advantages of this paper lie in its proposal of a lightweight anchor-based incremental framework capable of handling incremental multi-view data. By improving anchor generation and alignment methods and designing an efficient iterative convergence algorithm, it addresses the limitations of existing methods regarding the assumption of availability of all views and insufficient interaction in anchor generation among views. The proposed method demonstrates significant advantages in terms of time and space complexity and exhibits superior performance in extensive experiments.

**Strengths:**

The paper introduces a lightweight anchor-based incremental framework for multi-view clustering (LAIMVC) that effectively handles incremental multi-view data. This method improves anchor generation and alignment methods and designs an efficient iterative convergence algorithm, addressing the limitations of existing methods related to the assumption of all views being available and insufficient interaction in anchor generation among views. Specifically, LAIMVC demonstrates significant advantages in terms of time and space complexity, making it capable of handling sequentially arriving view data under limited storage and privacy constraints. It also shows outstanding performance on large-scale datasets. Extensive experiments have validated its effectiveness and practicality, showcasing excellent clustering performance, running time, and space efficiency.

**Limitations:**

Despite LAIMVC's excellent performance in handling incremental multi-view data, it has some limitations. First, the method relies on the initial selection and generation of anchors, and poor initial anchor selection might affect the final clustering results. Second, in certain complex data scenarios, LAIMVC's performance may be limited, requiring further optimization and parameter adjustment. Additionally, while the method has demonstrated superior performance in experiments, its real-world application needs more validation to ensure robustness and applicability across different datasets and application scenarios. Lastly, LAIMVC primarily addresses scenarios with incremental views, and further research and extension are needed to handle incremental samples or incomplete data effectively. Details as follows:

1、The introduction gives a good overview of multi-view clustering and mentions various methods. However, it lacks a specific problem statement that clearly outlines the unique challenge that this paper addresses.

2、Some technical terms and components of the proposed framework are introduced without sufficient explanation (e.g., the role of the permutation matrix in the algorithm).

3、The methodology section is densely packed with technical descriptions that might be challenging for readers to follow.

4、The experiments section includes extensive results but lacks detailed discussion on why certain datasets were chosen and the implications of the results.

5、The paper presents performance metrics but does not discuss the statistical significance of the results.

6、The conclusion summarizes the work done but does not adequately highlight the main findings or their implications.

**Suitability:**

2

---

### Official Review · Reviewer_THKp · 2024-05-31

**Rating:** 2
**Confidence:** 3

**Summary:**

This submission proposed a Lightweight Anchor-Based Incremental framework (LAIMVC) to address several issues in practical multi-view clustering such as the one-pass problem. The authors designed k-means-based initialization and permutation-matrix-based update for the anchor graph. Optimization and algorithm were presented in detail together with extensive experimental results.

**Strengths:**

This manuscript presents an anchor-based approach that deviates from conventional methods by employing anchor graphs to multi-view data in an ingenious way not explored before in this one-pass multi-view problem. The work is grounded in a mathematically elucidated problem formulation. The optimization and algorithm are sound, with detailed derivations and steps. The evaluation is comprehensive, benchmarking the approach against multiple baselines across diverse datasets.

**Limitations:**

1) Some one-pass multi-view works are missing. The authors are encouraged to clarify where the difference lies between one-pass multi-view clustering and continual multi-view clustering since they are facing the same problem of new views arriving sequentially over time.

[Ref.1] "One-pass multi-view learning." Asian conference on machine learning. PMLR, 2016.

[Ref. 2] "Online multi-view clustering with incomplete views." 2016 IEEE International conference on big data (Big Data). IEEE, 2016.

[Ref. 3] "One-pass incomplete multi-view clustering." Proceedings of the AAAI conference on artificial intelligence. Vol. 33. No. 01. 2019.

[Ref. 4] "Efficient one-pass multi-view subspace clustering with consensus anchors." Proceedings of the AAAI Conference on Artificial Intelligence. Vol. 36. No. 7. 2022.

[Ref. 5] "One-pass View-unaligned Clustering." IEEE Transactions on Multimedia (2024).

2) Some anchor-based multi-view works are missing. The authors are encouraged to clarify any novel contributions to the usage of anchor graphs in this submission. It seems your contribution only lies in applying to a new multi-view sub-scenario.

[Ref. 6] "Anchors bring ease: An embarrassingly simple approach to partial multi-view clustering." Proceedings of the AAAI conference on artificial intelligence. Vol. 33. No. 01. 2019.

[Ref. 7] "Adaptive anchor-based partial multiview clustering." IEEE Access 8 (2020): 175150-175159.

[Ref. 8] "Anchor-based incomplete multi-view spectral clustering." Neurocomputing 514 (2022): 526-538.

[Ref. 9] "Multi-View Fuzzy Clustering Based on Anchor Graph." IEEE Transactions on Fuzzy Systems (2023).

3) In the formulation step, the authors claimed to use PCA to deduct the feature dimensions of all views to a unified dimension because the inconsistent dimensions among views might lead to difficulty in anchor graph integration. The reviewer is confused as anchor graphs are a set of relationships between all samples and anchors. There is no original feature in the anchor graph, so how could it influence the anchor graph integration? Additionally, why do you use PCA rather than other more effective unsupervised algorithms? Also, there are too many missing details, for example, what does $B_t$ stand for? If it is for anchors, then why are there orthogonal constraints on anchor graph $Z_t$ and also $B_t$?

4) The details of features in each view of used datasets are missing. Additionally, the reviewer is curious if there are any GNN-based features used here that outperform hand-crafted features since the sample-to-sample relations can be incorporated before the anchor graph integration.

5) The proposed method has some randomness such as the initialization of means. Hence, all the results should be in the form of mean+/-standard deviation. It is not convincing based on the current results. Besides, it is recommended to do a significant test.

6) Did the baseline methods declare the parameters on all your datasets used here? Obviously not, but why do you directly use the recommendations on parameters rather than fine-tuning them? The reviewer is not convinced by your results from the baseline methods as they were not achieved with optimal parameters. Furthermore, could you explain how you fine-tuned the parameters in the experiments including using your unsupervised method?

7) Is there any difference if we re-arrange the orders of all views' sequences? Are there any results that reveal any insight?

8) Your convergence analysis and figures only show a local convergence. Is there any theoretical support for a global convergence?

**Suitability:**

3

---

### Meta-Review · Area_Chair_J51f · 2024-06-28

**Recommendation:** Accept (Poster)
**Confidence:** 5

**Metareview:**

This paper works lightweight anchor-based incremental framework for multi-view clustering with one-pass learning. The authors consider the anchor interaction across different views and alignment with graphs. They initialize an anchor graph with 𝑘-means for new views. The consensus anchor graph is updated with a permutation matrix. A three-step iterative learning algorithm is conceived for optimization. Experimental results show the effectiveness of the proposed method.

Four reviewers checked this paper, and one reviewer insisted on his/her opinion with a clear negative recommendation. The reviewer has concerns about the technical parts, like weak comparison and discussion with one-pass learning, anchor learning, anchor graph integration, and other detailed experimental settings. The overall idea of this paper is incremental to this research field. We have to reject this paper and its current form is not ready for ACM MM 2024.

***TPC Addendum ***
The paper has a clear split in opinions. Given the variation, and strong positive ratings, we would like to give the authors a chance to share their perspective at the conference.